# ACE Inhibitory Peptide from Skin Collagen Hydrolysate of *Takifugu bimaculatus* as Potential for Protecting HUVECs Injury

**DOI:** 10.3390/md19120655

**Published:** 2021-11-24

**Authors:** Shuilin Cai, Nan Pan, Min Xu, Yongchang Su, Kun Qiao, Bei Chen, Bingde Zheng, Meitian Xiao, Zhiyu Liu

**Affiliations:** 1College of Chemical Engineering, Huaqiao University, Xiamen 361021, China; caishuilin@hqu.edu.cn (S.C.); suyongchang@stu.hqu.edu.cn (Y.S.); 2Key Laboratory of Cultivation and High-value Utilization of Marine Organisms in Fujian Province, Fisheries Research Institute of Fujian, Xiamen 361013, China; npan01@qub.ac.uk (N.P.); xumin@jmu.edu.cn (M.X.); qiaokun@xmu.edu.cn (K.Q.); chenbei@fafu.edu.cn (B.C.); 3College of Food and Biological Engineering, Jimei University, Xiamen 361021, China

**Keywords:** *Takifugu bimaculatus*, skin collagen, HUVECs, antihypertension

## Abstract

Angiotensin-I-converting enzyme (ACE) is a crucial enzyme or receptor that catalyzes the generation of potent vasopressor angiotensin II (Ang II). ACE inhibitory peptides from fish showed effective ACE inhibitory activity. In this study, we reported an ACE inhibitory peptide from *Takifugu bimaculatus (T. bimaculatus)*, which was obtained by molecular docking with acid-soluble collagen (ASC) hydrolysate of *T. bimaculatus*. The antihypertensive effects and potential mechanism were conducted using Ang-II-induced human umbilical vein endothelial cells (HUVECs) as a model. The results showed that FNLRMQ alleviated the viability and facilitated apoptosis of Ang-II-induced HUVECs. Further research suggested that FNLRMQ may protect Ang-II-induced endothelial injury by regulating Nrf2/HO-1 and PI3K/Akt/eNOS signaling pathways. This study, herein, reveals that collagen peptide FNLRMQ could be used as a potential candidate compound for antihypertensive treatment, and could provide scientific evidence for the high-value utilization of marine resources including *T. bimaculatus*.

## 1. Introduction

Hypertension is ubiquitously considered a leading cause of cardiovascular morbidity and mortality [1,2]. It is generally believed that hypertension is closely related to the development of vascular endothelial dysfunction (VED), which is regarded as an early and changeable predictor in many cardiovascular pathological processes [3,4,5,6]. What is more, VED is a pathological phenomenon, which is characterized as reduced bioavailability of nitric oxide (NO) and/or by augmentation of reactive oxygen species (ROS), resulting in decreased endothelium-dependent vasodilatation and, consequently, leading to elevation of blood pressure [1,6,7]. Although the underlying mechanism of VED is complex and multifactorial, a large amount of evidence shows that oxidative stress is the main stimulating factor in this situation [5,7,8,9,10]. Angiotensin II (Ang II) is one of the strongest known vasoconstrictors in the renin-angiotensin system and is considered to be the key component of VED. In addition, Ang II directly stimulates reversible apoptosis of vascular endothelial cells through enhancing oxidative stress and inflammation [11,12]. Ang-II-induced apoptosis correlates with overproduction of ROS, accumulation of the potent vasoconstrictor, ET-1, and reduced endothelial nitric oxide synthase (eNOS) expression. It means that reducing ROS production and restoring NO synthesis might be an effective strategy against endothelial injury induced by Ang II [13,14,15]. Numerous studies have suggested that aside from being a successful therapeutic approach for hypertension, angiotensin-I-converting enzyme (ACE) inhibitors (ACEI) (e.g., Captopril (Cap) and Ramipril) could also improve endothelial function [16,17,18,19]. However, as we all know, these commercial antihypertensive agents also have some unpleasant side effects, such as dry cough, rashes, taste disorder, and angioneurotic edema [20]. Therefore, a great deal of attention has been paid to alternative ACEI, especially ACE inhibitory peptides derived from natural resources, due to the advantages of high bioavailability, excellent specificity, and low toxicity [21]. Marine organisms are excellent sources of bioactive natural compounds with various pharmacological effects (e.g., antihypertensive, antimicrobial, and immunomodulatory). They are considered the largest and most valuable remaining repository of novel molecules [22]. In recent years, more and more researchers have paid attention to marine bioactive peptides [23]. Up to date, extensive ACE inhibitory peptides have been identified from marine fish, shellfish, shrimp, and algae, indicating that marine-derived proteins may become potential sources of antihypertensive peptides [22,24,25,26].

Pufferfish, also known as fugu, belong to the family Tetraodontidae and the order Puffiniformes, and are a notable toxic fish species distributed in the coastal and offshore areas of China [27]. Considering that they contain deadly tetrodotoxin, China has banned the sale and consumption of pufferfish [27]. Thanks to a series of technological breakthroughs by domestic and foreign researchers in recent years, the production of non-toxic pufferfish (e.g., *Takifugu obscurus* and *Takifugu bimaculatus*) has been realized. China is the largest producer of pufferfish, and plenty of by-products are produced during fish processing, including fish skin, fish livers, and fishbone. Among them, fish skin is the main by-product, accounting for about 27% of the pufferfish’s total weight (wet basis). It is rich in collagen and has a high utilization value. Many studies have shown that marine collagen peptide has a variety of biological activities, including antibacterial, antioxidant, immunomodulatory, and antihypertensive activities [28]. As a significant commercially farmed species, the current *Takifugu bimaculatus* (*T. bimaculatus*) research mainly focuses on toxicity, breeding, and nutritional evaluation [29,30,31]. To date, many new ACE inhibitory peptides have been identified from marine resources, especially from the skin of marine fishes [24,32]. However, to the best of our knowledge, biological activities involving antihypertensive mechanisms and effects of *T. bimaculatus* have not yet been reported.

Based on this, herein, this article aimed to screen and identify *T. bimaculatus* collagen peptide (FNLRMQ) by enzymatic hydrolysis, ultrafiltration, and molecular docking, and to evaluate the ACE inhibitory activities by in vitro assay. We also used Ang-II-stimulated human umbilical vein endothelial cells (HUVECs) as an endothelial injury model, and studied their effects on apoptosis and oxidative stress. The results showed that FNLRMQ has a good antihypertensive effect, which indicates that FNLRMQ could be used as a potential candidate compound for antihypertensive treatment, and provides a scientific basis for the high-value utilization of marine resources, including *T. bimaculatus*.

## 2. Results and Discussion

### 2.1. Screening and Identification of ACE Inhibitory Peptides

ACE inhibitory peptides are encrypted within the protein’s primary structure, where they remain inactive before being released by thermal or enzymatic hydrolysis [33]. In this study, ACE inhibitory peptides were screened and identified from acid-soluble collagen (ASC) of *T. bimaculatus* by enzymatic hydrolysis combined with ultrafiltration, LC-MS, and molecular docking. In brief, the collagen was hydrolyzed by alkaline protease and then classified into three different molecular weight (MW) fractions through 1 kDa and 5 kDa ultrafiltration membranes, which were labeled as TBSH-I (MW < 1 kDa), TBSH-II (1 kDa < MW < 5 kDa), and TBSH-III (MW > 5 kDa). The fraction with the highest ACE inhibitory activity (TBSH-I, Appendix A) was selected for further LC-MS/MS analysis. Using PEAKS Studio, based on their de novo sequence characteristics and the average local confidence yield of 95%, a total of 82 peptide sequences (Appendix A) were first selected. Subsequently, molecular docking was conducted by MOE software. The interaction between peptides and ACE was evaluated based on the scoring function of MOE-Dock (Appendix A). In this study, peptides with a score of < –15 kcal/mol were selected for chemical synthesis. The peptide’s purity was examined by HPLC and MS (Appendix A,S3), followed by ACE inhibitory activity validation, and FNLRMQ exhibited the highest ACE inhibitory activity of 80.35% (Appendix A). To better understand the interaction between FNLRMQ and ACE, binding-mode analysis between FNLRMQ with ACE was performed. The contact list between FNLRMQ and ACE is shown in Appendix A.

As illustrated in Appendix A, the nitrogen atom of the imidazole group of H353 in ACE, forms one hydrogen bond with the sulfur atom of the thioether group of M5 in FNLRMQ. The oxygen atom of the carboxyl group of D453 in ACE, forms one hydrogen bond with the nitrogen atom of the backbone of F1 in FNLRMQ. The nitrogen atom of the amino group of K454 in ACE, forms one hydrogen bond with the oxygen atom of the amido group of N2 in FNLRMQ. The oxygen atom of the phenolic hydroxyl group of Y523 in ACE, forms one hydrogen bond with the oxygen atom of the amido group of Q6 in FNLRMQ. The oxygen atom of the carboxyl group of E162 in ACE, forms salt bridges with the nitrogen atoms of the guanidine group of R4 in FNLRMQ. The oxygen atom of the carboxyl group of D376 in ACE, forms a salt bridge with the nitrogen atom of the guanidine group of R4 in FNLRMQ. The oxygen atom of the carboxyl group of E377 in ACE, forms a salt bridge with the nitrogen atom of the guanidine group of R4 in FNLRMQ. The Zn^2+^ in ACE, forms an ion contact with the oxygen atom of the amido group of Q6 in FNLRMQ. Docking simulation studies indicate that Zn^2+^, E162, H353, E376, D377, D453, K454, and Y523, the residues in ACE, are involved in binding with F1, N2, R4, M5, and Q6, the residues in FNLRMQ, through ion contact, salt bridges, and hydrogen-bond interactions.

Finally, FNLRMQ was submitted to the antihypertensive peptides database (AHTPDB, http://crdd.osdd.net/raghava/ahtpdb, accessed on 23 November 2021) and the active peptide database (BIOPEP, http://www.uwm.edu.pl/biochemia/biopep/start_biopep.php, accessed on 23 November 2021). Surprisingly, no matching data was found, indicating that FNLRMQ was a novel antihypertensive peptide (Phe-Asn-Leu-Arg-Met-Gln, Figure 1a). Therefore, FNLRMQ was selected for further investigation to determine its antihypertensive capacity and potential mechanism.

### 2.2. FNLRMQ Enhanced Cell Viability in Ang-II-induced HUVECs

The potential cytotoxicity of FNLRMQ with different concentrations of HUVECs and the protective effect of FNLRMQ in Ang-II-induced HUVECs were assessed by CCK-8 assay after incubation for 24 h. As presented in Figure 1b, HUVECs’ viability was not affected by FNLRMQ (1–40 µM) within 24 h incubation. It is reported that Cap could inhibit the secretion of endothelin-1 and increase the production of NO at a concentration of 0.1 µM [34]. Therefore, we adopted Cap as a positive control and evaluated the effect on HUVECs’ viability. As expected, Cap showed no significant impact on HUVECs’ viability at the concentration from 1 nM to 500 nM (Figure 1c).

As shown in Figure 1d, after being treated with Ang II, cell viability decreased considerably compared with the normal group, and FNLRMQ effectively attenuated the proliferation inhibition in a dose-dependent manner. Both FNLRMQ (5–20 µM) and Cap increased cell viability relative to that of the Ang-II-induced model group to some extent. Thus, FNLRMQ alone does not affect cell viability, and FNLRMQ treatment protects HUVECs from Ang-II-stimulated viability reductions.

### 2.3. FNLRMQ Ameliorated Ang-II-Induced HUVEC Apoptosis

The functional integrity of endothelial cells plays an essential role in the health of arteries. Extensive research has shown that endothelial dysfunction is an initiator and a critical factor in developing cardiovascular disease [35]. Endothelial cell dysfunction can be triggered by various stimulations (e.g., oxidative stress, chronic inflammation, infection, Vitamin D deficiency, and shear stress). Experimental studies have established that Ang II promotes apoptosis of HUVECs by producing an abundance of ROS [36,37,38].

Apart from being used in the medical treatment of cardiovascular diseases, ACEI also exhibits immune-regulating, antioxidant, and anti-inflammatory properties [38]. It has been reported that Cap can reverse oxidative stress and subsequent apoptosis of human coronary artery endothelial cells (HCAECs) [39,40]. Chen et al. reported that an ACE inhibitory peptide alleviates Ang-II-induced apoptosis in HUVECs [41]. Therefore, we hypothesized that ACE inhibitory peptide (FNLRMQ) could ameliorate endothelial dysfunction and suppress apoptosis of HUVECs exposed to Ang II treatment.

The flow cytometry assay was performed to investigate whether reductions in cell viability following Ang II treatment were due to increased HUVEC apoptosis. To evaluate the capacity of FNLRMQ to alleviate Ang-II-induced apoptosis in HUVECs, expression of apoptosis-regulatory proteins Bax, Bcl-2, activated caspase-3, and cytochrome C was assessed by Western blotting. Percentages of apoptotic cells were dramatically increased in the Ang II exposure group compared with normal control (Figure 2a,b). Interestingly, both Cap and FNLRMQ treatment significantly alleviated Ang-II-induced HUVEC apoptosis. FNLRMQ treatment (5–20 µM) decreased the apoptosis rate in a dose-dependent manner, and the apoptosis rate of HUVECs decreased to 22.44%, 13.25%, and 9.49%, respectively. There are increasing pieces of evidence that higher levels of the Bax/Bcl-2 ratio may lead to cell susceptibility to apoptosis [42]. Thus, the expression of the Bcl-2 and Bax was also determined by Western blotting analysis. As shown in Figure 2c,d, stimulation of Ang II caused a significant augmentation of Bax and c-caspase-3, together with decreased expression of the anti-apoptotic protein, Bcl-2, compared with normal control. Compared with the model control cells, the Bax/Bcl-2 ratio decreased significantly by 29.47%, 48.98%, and 62.51% after treatment with FNLRMQ at 5 µM, 10 µM, and 20 µM, respectively (Figure 2e). The spontaneous release of cytochrome C from mitochondria to the cytosol is associated with subsequent activation of caspase-3 for apoptosis [43]. A remarkable increase in expression of cytochrome C was observed in the Ang-II-treated group, which was then reversed by Cap and FNLRMQ treatment. In summary, FNLRMQ may reduce the apoptosis of HUVECs by down-regulating the expression of Bax, caspase-3, and cytochrome C and up-regulating the expression of Bcl-2, thus significantly improve the cell injury induced by Ang II.

### 2.4. Protective Effects of FNLRMQ on Oxidative Stress in Ang-II-Induced HUVECs

It is generally accepted that oxidative stress damages proteins, lipids, and DNA, promoting endothelial dysfunction and subsequent pathophysiological events [44]. ROS plays a vital role as a regulator of cell survival, cell death, cell signaling, and inflammation [45], and excessive production of ROS will lead to oxidative stress. More and more evidence shows that the generation and accumulation of ROS promotes apoptosis of blood vessels and neurons [46]. Ang-II-induced HUVEC dysfunction may also be accompanied by various damage markers, such as decreased SOD and CAT activities and increased ROS and MDA levels [47,48]. Consistent with these findings, we noticed that the FNLRMQ- and Cap-treatment groups exhibited a significant decrease in the levels of ROS (Figure 3a,b), MDA (Figure 3c), and LDH (Figure 3d), as well as an increase in T-AOC (Figure 3e). Based on the above results, we believed that FNLRMQ protected against Ang-II-induced oxidative stress and inhibited the apoptosis of HUVECs by improving the antioxidant capacity.

### 2.5. FNLRMQ Facilitated the Translocation of Nrf2 and Activated the Nrf2/HO-1 Pathway

Nrf2 is the most prominent regulator of cell resistance to oxidants and mediates the antioxidant response to inflammation and oxidative stress. Hence, we investigated the Nrf2 signaling pathway to elucidate the antioxidant mechanisms of FNLRMQ against Ang-II-induced oxidative stress. The expression of Nrf2 in the nucleus and cytoplasm of HUVECs was determined by Western blotting analysis (Figure 4a–c) and immunofluorescence (IF) technique (Figure 4d,e). Results revealed that the Nrf2 level in the cytoplasm was slightly decreased in the Ang-II-induced model group, and significantly decreased in the 20 µM-of-FNLRMQ treatment group, compared with the normal control group. Conversely, the expression of Nrf2 in the nucleus was slightly increased in Ang-II-treated cells. Compared with Ang-II-treated cells, 10 µM and 20 µM of FNLRMQ considerably promoted this increase. The results of IF were similar to those of Western blotting analysis.

To evaluate the involvement of Nrf2-dependent antioxidant responses of cellular resistance to Ang II, the expression of downstream effector HO-1 (Figure 4f,g) was measured through Western blotting analysis, and the levels of SOD and GSH were also detected (Figure 4h,i). For the expression of HO-1, only the 10 µM and 20 µM of FNLRMQ exhibited considerable difference, compared with the Ang-II-induced model group. However, the expression of Keap-1 did not change significantly. As expected, the activities of SOD and GSH were decreased with Ang II stimulation, and FNLRMQ restored the levels of SOD and GSH in a dose-dependent manner.

To our knowledge, vascular oxidative stress has a prominent place in the pathologic process of cardiovascular diseases [44,49,50]. The leading cause for the majority of pathophysiological conditions is inadequate protection against oxidative-stress-induced cellular injury. Many antioxidant signaling pathways are involved in the regulation of ROS balance. Particularly, the Nrf2/Keap1-antioxidant response element signaling pathway may be the main one. Previous studies on Ang II have manifested that the Nrf2 pathway is involved in the promotion of oxidative stress and the production of proapoptotic proteins in injured HUVECs [51,52,53]. The overall implication from the above results is that FNLRMQ protects HUVECs against cell injury by promoting the levels of nuclear Nrf2, thereby activating the Nrf2/HO-1 pathway.

### 2.6. The Effects of FNLRMQ on the Production of NO and NOS

The NO generated by eNOS is the central regulator of vasodilation [54], and there is an increasing awareness that insufficient NO bioavailability may contribute to endothelial dysfunction [55]. Studies have confirmed that Ang II downregulates phosphorylated eNOS to reduce NO production [56,57]. Production of NO and activity of NOS were assessed in Ang-II-stimulated HUVECs to elucidate any role that FNLRMQ might have in protection against cellular damage. As presented in Figure 5a,b, the production of NO and the activity of NOS in Ang-II-induced groups was decreased significantly compared with the normal group. Notably, both Cap and FNLRMQ treatment productively facilitated NO and NOS generation. These results suggested that FNLRMQ treatment was able to suppress the degree of endothelial dysfunction induced by Ang II.

### 2.7. The Role of the eNOS in Response to Ang-II-Induced Oxidative Stress

As discussed above, maintenance of redox homeostasis and inhibition of excessive ROS generation would seem to be an effective approach to protect endothelial cells from injury.

ROS-producing systems in the vascular wall are dominated by NADPH oxidase (NOX), xanthine oxidase (XO), and uncoupled nitric oxide synthase (NOS) and the uncoupling of eNOS contributes markedly to increased ROS production [58]. eNOS is a downstream target of activated Akt and the foremost and constitutionally expressed NOS in vascular endothelial cells, accelerating the NO synthesis. eNOS was phosphorylated by Akt in response to a variety of cellular stimuli [59].

Our findings demonstrated that FNLRMQ facilitated NO synthesis and suppressed ROS accumulation, indicating a putative role for FNLRMQ in relieving Ang-II-induced damage. Thus, we hypothesized that the Akt/eNOS pathway might be responsible for the dysfunction resulting from Ang-II-induced oxidative stress, including reduced NO levels.

Therefore, Western blotting analysis was conducted to investigate the effect of eNOS against Ang-II-treated HUVECs. As presented in Figure 5c,d, the addition of Ang II suppressed the total eNOS and phosphorylated eNOS (p-eNOS) compared to the normal group. Treatment with either 10 µM or 20 µM FNLRMQ induced a significant augmentation in both eNOS and p-eNOS, while treatment with 5 µM FNLRMQ had no effect on eNOS and p-eNOS expression, compared with the Ang-II-induced model group. A specific inhibitor of eNOS, L-NAME, was used to further explore the involvement of eNOS in FNLRMQ action against Ang-II-induced oxidative stress. HUVECs were pre-incubated with 100 μM of L-NAME and then subjected to Ang II, FNLRMQ + Ang II, or Cap + Ang II. NO production was measured as previously described. As presented in Figure 5e, L-NAME reduced Cap, and FNLRMQ improved NO production. Another study also showed a similar finding [60,61]. These results revealed that FNLRMQ had its effect on Ang-II-induced cell dysfunction by activating the Akt/eNOS pathway, leading to NO production.

### 2.8. FNLRMQ Attenuated Ang-II-Induced HUVECs Dysfunction via PI3K/Akt Pathway

Growing evidence suggests that both eNOS and Nrf2 are regulated through PI3K/Akt, which plays a principal role in biological homeostasis in response to oxidative stress and inflammation [62,63,64]. Additionally, PI3K/Akt promoted the nuclear translocation and transactivation activity of Nrf2 [65]. Based on the results outlined above, we speculated that FNLRMQ protects HUVECs against Ang-II-induced dysfunction via the PI3K/Akt pathway. More specifically, we considered that the Akt/eNOS and Nrf2/HO-1 pathways were involved in the FNLRMQ protection of cell damage to HUVECs via the regulation of the PI3K/Akt pathway. In order to confirm our suspicions, we determined the expression of PI3K, Akt, and phosphorylated Akt after Ang II stimulation with or without FNLRMQ.

As presented in Figure 6a,b, the Ang-II-treated group decreased the expression of PI3K. Moreover, Ang-II-treated HUVECs inhibited p-Akt expression while the total Akt expression exhibited no difference compared to the normal group. It should be noted that FNLRMQ treatment significantly increased the expression and phosphorylation of Akt in a dose-dependent manner. Additionally, FNLRMQ-treated HUVECs showed recovery of PI3K expression, with significant differences between 10 μM and 20 μM groups only.

To investigate the involvement of PI3K in FNLRMQ protection against Ang-II-induced HUVEC dysfunction, experiments were performed using the PI3K inhibitor, LY294002. HUVECs were pre-incubated with 50 μM LY294002 for 30 min, followed by treatment with Ang II for 24 h in the absence or presence of FNLRMQ (20 μM). Levels of ROS and NO plus cytosolic and nuclear Nrf2 were determined.

The results suggested that Ang II raised the expression of Nrf2 in the nucleus, and both FNLRMQ and Cap treatment enhanced this expression (Figure 6c–e). LY294002, however, inhibited the expression of Nrf2 in the nucleus. The PI3K/Akt/eNOS pathway mediates NO formation via promoting phosphorylation of eNOS under all manner of stimuli [66]. Similarly, we explored whether the PI3K regulated the synthesis of NO (Figure 6g). As shown in Figure 6g, LY294002 decreased the restoration of NO production resulting from Cap or FNLRMQ treatment. LY294002 abolished the reduction in ROS resulting from FNLRMQ or Cap treatment (Figure 6f). The results presented above substantiate the view that the protective effect against Ang-II-induced HUVEC dysfunction evinced by FNLRMQ is achieved via the Nrf2/HO-1 and PI3K/Akt/eNOS signaling pathways.

## 3. Materials and Methods

### 3.1. Materials and Reagents

Peptides were synthesized at GenScript (Nanjing, China). Human Ang II and Cap were obtained from Yuanye Bio-Technology Co., Ltd. (Shanghai, China). Dulbecco’s Modified Eagle Medium/Nutrient Mixture F-12 (DMEM/F12), fetal bovine serum (FBS), phosphate-buffered saline (PBS), trypsin (0.25%), and penicillin–streptomycin solution were obtained from Fisher Scientific (Waltham, MA, USA). The antibodies used in Western blot analysis, including Bax, active-caspase-3, Nrf2, H3, keap1, GSH, HO-1, p-NOS, eNOS, PI3K, Akt, and p-Akt were acquired from Abcam (Cambridge, UK). Bcl-2 and SOD antibodies were purchased from Proteintech Group (Rosemont, IL, USA). GAPDH was purchased from Cell Signaling (Beverly, MA, USA). All other reagents (analytical grade) were purchased from Xiamen LvYin Reagent and Glass Apparatus Co., Ltd. (Xiamen, China).

### 3.2. Production of Collagen Hydrolysates

The ASC of *T. bimaculatus* was dispersed in distilled water to obtain a protein suspension (10%, *w*/*v*), and then the pH was adjusted to 12. The alkaline protease was added to the suspension at an enzyme-to-substrate (E/S) ratio of 2% (*w*/*w*). The hydrolysis was carried out at 55 °C for 6 h. The reaction was terminated in a boiling water bath at 100 °C for 10 min. Before further analysis, the hydrolysate was freeze-dried and stored at –20 °C.

### 3.3. Fractionation by Ultrafiltration

The ASC hydrolysates were fractionated through two different UF membranes (1 kDa and 5 kDa) in roll ultrafiltration membrane equipment (Shaoxing Heiner Membrane Technology Co., Ltd.) with a range of MW cut-offs from 1 k to 5 kDa permeate and retentate at each stage of filtration was collected, and the obtained fractions were noted as TBSH-I (MW < 1 kDa), TBSH-II (1 kDa < MW < 5 kDa), or TBSH-III (MW > 5 kDa). The three fractions were freeze-dried and then analyzed for ACE inhibitory activity in vitro.

### 3.4. Screening and Identification of the Potential Bioactive Peptides

LC-MS/MS analysis of the peptide fractions with the best ACEI capacity was carried out by an Ultimate 3000 UHPLC (Thermo Fisher, Waltham, MA, USA) connected to Q-Exactive Mass Spectrometers (Thermo Fisher, Waltham, MA, USA). Both MS spectra and MS/MS spectra were recorded in reflector mode within a mass/charge (*m/z*) range of 350–1800. Operating conditions of the MS analysis were as follows: resolution = 70,000, AGC target = 3 × 10^6^, and maximum IT = 40 ms. Operating conditions of the MS/MS analysis were as follows: resolution = 17,500, AGC target = 1 × 10^5^, maximum IT = 60 ms, topN = 20, and NCE/stepped NCE = 27 kV. The error of MS and MS/MS was < 5 and 10 ppm, respectively. The spectral raw files were analyzed by Peaks Studio (Bioinformatics Solutions Inc.; Waterloo, ON, Canada) to identify the amino sequences of potential bioactive peptides. Subsequently, molecular docking between these peptides and ACE was conducted by MOE-Dock. The peptides as a ligand and ACE as a receptor and the 3D structure of and active sites of ACE (Appendix A) were downloaded from the RCSB Protein Data Bank (PDB ID: 1O8A) and InterPro (https://www.ebi.ac.uk/interpro/, accessed on 23 November 2021), respectively. MOE software was used to prepare docking by minimizing their energy and then 3D protonating [67]. The binding site of the native ligand in the ACE protein structure was set as the binding pocket for peptides (Appendix A). All docked poses of peptides were ranked by London dG scoring. The peptides with a < –15 kcal/mol docking score were selected for chemical synthesis and ACE-inhibitory-activity validation.

### 3.5. Determination of ACE Inhibitory activity

The ACE inhibitory activity was determined using HHL (5 mM) as Ma et al. described with slight modifications [68]. Both HHL and ACE solutions were prepared in 0.1 M borate buffer containing 0.3 M NaCl (pH = 8.3). Briefly, sample solutions (50 μL) were added into 150 μL of 5 mM HHL solution and then pre-incubated at 37 °C for 5 min. Subsequently, the ACE solution (50 μL, 0.05 U/mL) was added to the mixture, then incubated at 37 °C for 45 min. The enzymatic reaction was terminated by the addition of 200 μL of 1.0 M HCl, and the solution was filtered through a 0.22 μm nylon syringe filter before RP-HPLC analysis (Waters, Milford, MA, USA) on a SunFire C18 Column (100 Å, 5 µm, 4.6 mm x 150 mm). The mobile phase consisted of distilled water and acetonitrile (68:32, *v*/*v*, with 0.1% TFA). The flow rate was set at 1 mL/min, and the UV detection was carried out at a wavelength of 228 nm. The inhibitory activity was calculated using the equation as follows:
$$\mathrm{ACE}\;\mathrm{inhibitory}\;\mathrm{activity}\;\left(\%\right)=\frac{A-B}{A}\times 100$$ where A is the peak area of hippuric acid in the presence of both ACE and the sample, and B is the peak area of hippuric acid without a sample (buffer for samples).

### 3.6. Cell Culture

HUVECs were obtained from Meixuan Biological Science Co. Ltd. (Shanghai, China) and cultured in DMEM/F12 supplemented with 10% FBS (*v*/*v*), 100 μg/mL streptomycin, and 100 U/mL penicillin in a humidified incubator (Thermo, Waltham, MA, USA) with 5% CO_2_ at 37 °C. Experimental groups were designated as follows: (a) normal group: HUVECs without Ang II, FNLRMQ, or Cap treatment; (b) model group: HUVECs induced with Ang II; (c) protective group: HUVECs treated with Ang II and FNLRMQ; (d) positive control group: HUVECs treated with Ang II and Cap.

### 3.7. Cell Viability Assessment

The viability of HUVECs was evaluated by the CCK-8 assay kit (Beyotime, Shanghai, China), according to the manufacturer’s instructions. In brief, HUVECs were seeded in 96-well plates (2 × 10^4^ cells/well) and cultured in an incubator at 37 °C for 24 h. After cellular starvation with an FBS-free medium for 12 h, HUVECs were incubated at various experimental conditions. Subsequently, HUVECs were washed twice with PBS, then CCK-8 was added to each well plate and cultured for about 1 h. Absorbance at 490 nm was determined through a Thermo Multiskan Mk3 Microplate Reader (Waltham, MA, USA).

### 3.8. Apoptosis Analysis

Flow cytometry (BD, Franklin Lakes, NJ, USA) was adopted to analyze the impact of FNLRMQ on apoptosis induced by Ang II treatment. Briefly, HUVECs were plated in 6-well plates (5 × 10^4^ cells/well) for 24 h. After 12 h quiescence, HUVECs were incubated at various experimental conditions. The HUVEC culture solution was trypsinized, centrifuged, and then washed once with pre-cold PBS buffer and resuspended with dilution of binding buffer at a final concentration of 1–5 × 10^6^ cells /mL (1:3 dilution in ddH_2_O of 4× binding buffer). Then, 100 µL of HUVEC suspension was mixed with 5 µL annexin V/FITC and incubated in darkness for 5 min at 25 °C. Finally, 10 µL propidium iodide (PI, 20 µg/mL) plus 400 µL PBS buffer was added to perform flow cytometry analysis.

### 3.9. Detection of ROS and NO

2′,7′-dichloro-dihydro-fluorescein diacetate (DCFH-DA, Solarbio, Beijing, China) and 4-amino-5-methylamino-2′,7′-difluorofluorescein diacetate (DAF-FM DA, Meilunbio, Dalian, China) were used to measure intracellular ROS and NO production, respectively. After incubation with Ang II, FNLRMQ, or Cap, cells were incubated with diluted DCFH-DA (10 µM) and DAF-FM DA (5 µM) for 20–25 min at 37 °C. HUVECs were washed three times with PBS and measured by inverted fluorescence microscopy (Cewei, Shanghai, China) and Microplate Reader (MD, San Jose, CA, USA), respectively.

### 3.10. Biochemical Indicator Detection

HUVECs were seeded in 6-well plates (5 × 10^4^ cells/well) and induced with Ang II, Ang II + FNLRMQ, and Ang II + Cap for 24 h. For eNOS and PI3K inhibition, before being treated with Ang II, FNLRMQ, or Cap, HUVECs were pre-incubated with L-NAME (100 μM) and LY294002 (50 μM) for 30 min, respectively. After 24 h, the supernatants of HUVECs were collected. The activities or levels of MDA, SOD, LDH, T-AOC, and GSH were measured using assay kits, following the manufacturer’s instructions.

### 3.11. Subcellular Fractionation

The distribution of Nrf2 in the nucleus and cytosol was determined by a cytosolic and nuclear protein extraction kit (Beyotime, Shanghai, China). In brief, HUVECs were washed with PBS once, collected, and centrifuged at 3000 rpm for 10 min at 4 °C. Supernatants were discarded, and 200 μL of nuclear and cytosolic protein extraction solution A with 0.1 mM PMSF was added to 20 μL of precipitate and vortexed at maximum speed for 5 s. Cell extracts were placed in an ice bath for 15 min. Nuclear and cytosolic protein extraction solution B was added, and the suspension was centrifuged at 11,000 rpm for 5–10 min at 4 °C. The supernatant was stored as a cytoplasmic fraction at –80 °C. The precipitate was resuspended with 50 μL of nuclear and cytosolic protein extraction solution with 0.1 mM PMSF and vortexed for 30 s, and then placed in an ice bath for 2 min. This process was repeated several times over a 30 min duration. Finally, the samples were centrifuged at 11,000 rpm for 10–15 min at 4 °C. The supernatant was immediately aspirated into a pre-cooled plastic tube and frozen at –80 °C.

### 3.12. Western Blotting Analysis

HUVECs cultured with Ang II, Ang II + FNLRMQ, and Ang II + Cap were washed three times with a pre-cold PBS buffer. Cells were pretreated with a protease inhibitor (Thermo Scientific, Waltham, MA, USA) for 5 min before adding 0.1 mM of RIPA lysis buffer (Beyotime, Shanghai, China) containing PMSF. The levels of proteins were measured using a BCA protein assay kit (Pierce, Waltham, MA, USA). For Western blotting, 50 μg of total protein were separated by 12% SDS-PAGE and transferred to PVDF membranes blocked with 5% skimmed milk. The PVDF membranes were washed three times with TBST buffer for 5–10 min and probed with primary antibodies for about 2 h at 25 °C, followed by incubation with secondary antibody (goat anti-mouse or goat anti-rabbit antibody) for approximately 1 h. The PVDF membranes were washed 3–5 times with TBST; bands visualized utilizing an ECL Reagent kit (Invitrogen, Waltham, MA, USA) were then detected using an imaging system (Tanon Science and Technology Co., Ltd., Shanghai, China). The optical densities of target bands were analyzed with Image-Pro software.

### 3.13. Statistical Analysis

All results are presented as means ($\overset{-}{x}$) ± standard deviation (SD). Statistical analysis was carried out via SPSS Statistics software (Chicago, IL, USA). ANOVA plus Dunnett’s test and LSD were adopted to determine differences among multiple experimental groups. *p* < 0.05 was considered as statistical significance.

## 4. Conclusions

The present study identified a novel peptide sequence with ACE inhibitory activity from skin collagen hydrolysate of pufferfish (*T. bimaculatus*) by enzymatic hydrolysis, ultrafiltration, and molecular docking. Furthermore, the antihypertensive mechanism was studied using Ang-II-induced HUVECs as an endothelial injury model. The results suggested that FNLRMQ provides effective protection against Ang-II-induced HUVEC dysfunction. FNLRMQ treatment increased viability and attenuated apoptosis of HUVECs by regulating the expression of Bax, cleaved caspase-3, cytochrome C, and Bcl-2. FNLRMQ also increased NO production and decreased the augmentation of ROS to alleviate oxidative stress damage. These effects were mediated via the promotion of Akt and eNOS phosphorylation and translocation of Nrf2. The results confirmed that the protective function of FNLRMQ was regulated through Nrf2/HO-1 and PI3K/Akt/eNOS pathways. Thus, we conclude that FNLRMQ is a potential candidate for the therapy of cardiovascular diseases (e.g., hypertension).

## Figures and Tables

**Figure 1 marinedrugs-19-00655-f001:**
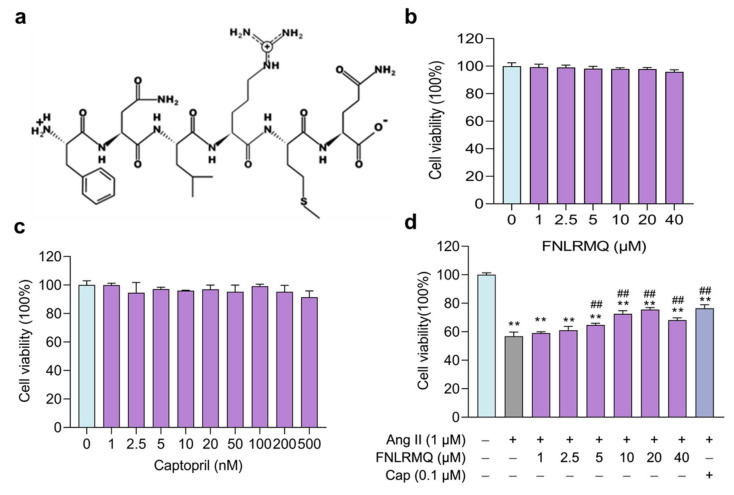
FNLRMQ enhanced cell viability in Ang-II-induced HUVECs. (**a**) Structure of FNLRMQ. (**b**) Evaluation of the effect of FNLRMQ against the HUVECs. HUVECs were treated with FNLRMQ (1–40 µM) without Ang II treatment for 24 h. (**c**) Evaluation of the effect of Cap against the HUVECs. HUVECs were treated with Cap (1–500 nM) without Ang II for 24 h. (**d**) Effect of FNLRMQ (1–40 µM) on the viability of Ang-II-induced HUVECs. Cap as the positive control. HUVECs’ viability was measured using a CCK-8 assay. The values are expressed as the means ± SD, n = 3. ** *p* < 0.01, ^##^
*p* < 0.01. The figure marked ** was compared with the normal control group and the figure marked ## was compared with the Ang-II-induced-model group.

**Figure 2 marinedrugs-19-00655-f002:**
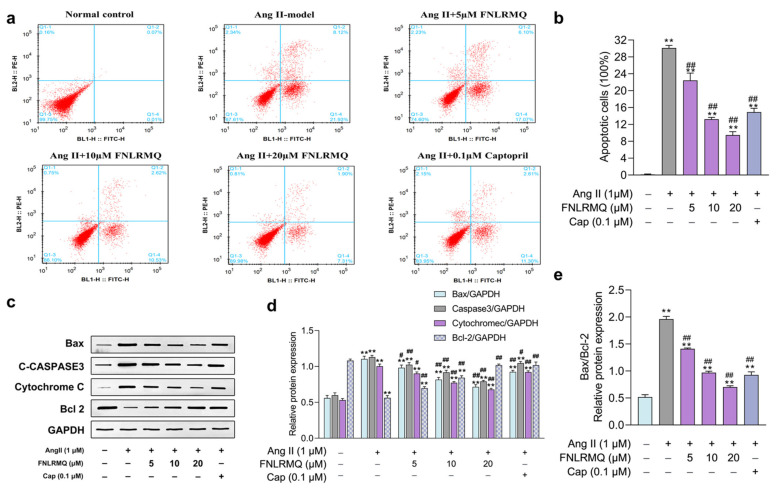
FNLRMQ ameliorated Ang-II-induced HUVECs apoptosis. HUVECs were incubated with Ang II for 24 h in the absence or presence of FNLRMQ (5–20 µM). (**a**) The flow cytometry assay was adopted to detect the effect of FNLRMQ on the apoptosis of Ang-II-induced HUVECs. (**b**) The percentage rate of apoptotic cells (mean ± SD, n = 3). (**c**) Expression of cytochrome C and anti-apoptotic protein Bcl-2 and pro-apoptosis proteins caspase-3 and Bax was examined by Western blotting analysis in HUVECs with different treatments. Glyceraldehyde 3-phosphate dehydrogenase (GAPDH) as an internal control. (**d**,**e**) The quantification of the immunoreactive bands. The values are expressed as the means ± SD, n = 3. ** *p* < 0.01, ^#^
*p* < 0.05, ^##^
*p* < 0.01. The figure marked ** was compared with the normal control group and the figure marked ## was compared with the Ang-II-induced-model group.

**Figure 3 marinedrugs-19-00655-f003:**
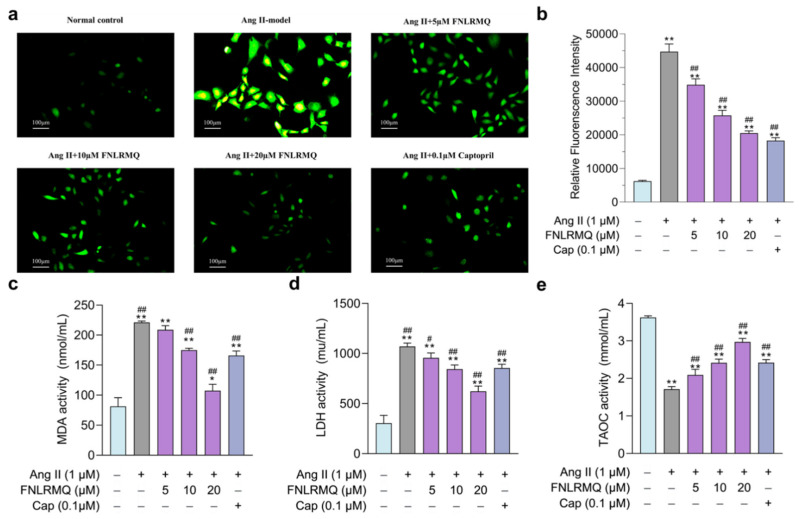
Protective effects of FNLRMQ on oxidative stress in Ang-II-induced HUVECs. HUVECs were incubated with Ang II for 24 h in the absence or presence of FNLRMQ (5–20 µM). (**a**) The fluorescent probe, DCFH-DA was adopted to measure total intracellular ROS level in HUVECs, and photos were taken by fluorescence inverted microscopy. (**b**) Mean fluorescence intensity was analyzed by ImageJ software. The values are expressed as the means ± SD, n = 3. (**c**,**d**) HUVECs’ oxidative damage was assessed by MDA and LDH detection. (**e**) T-AOC was determined using an assay kit. The values are expressed as the means ± SD, n = 3. ** *p* < 0.01, * *p* < 0.05, ^##^
*p* < 0.01, ^#^
*p* < 0.05. The figure marked ** was compared with the normal control group and the figure marked ## was compared with the Ang-II-induced-model group.

**Figure 4 marinedrugs-19-00655-f004:**
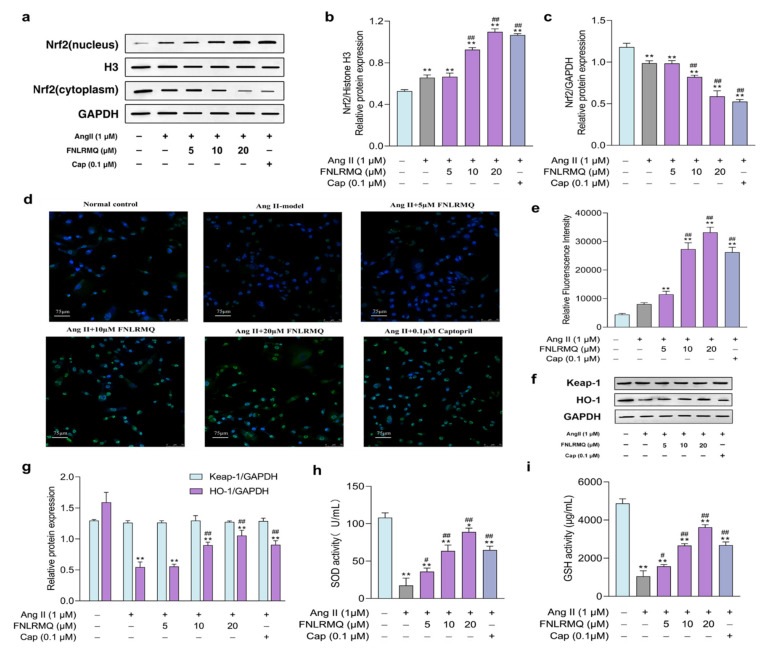
FNLRMQ catalyzed the Nrf2 nuclear translocation and activation of the Nrf2/HO-1 signaling pathway. HUVECs were incubated with Ang II for 24 h in the absence or presence of FNLRMQ (5–20 µM). (**a**) Effect of FNLRMQ on protein expression of Nrf2 in the nucleus and cytoplasm. Histone H3 and GAPDH were used as the internal control. (**b,c**) The quantification of the immunoreactive bands of Nrf2 in the nucleus and cytoplasm, respectively. (**d**) IF staining of HUVECs with anti-Nrf-2 antibody (green) and DAPI (blue). (**e**) Mean fluorescence intensity of IF. The values are expressed as the means ± SD, n = 3. (**f**) Effect of FNLRMQ on the expression of antioxidant enzyme in HUVECs. Protein expression of the HO-1 and Keap-1 was measured by Western blotting analysis. GAPDH functioned as the control. (**g**) The quantification of the immunoreactive bands. (**h**,**i**) The effects of FNLRMQ on the activities of antioxidant enzymes SOD and GSH were determined by kit assays. The values are expressed as the means ± SD, n = 3. ** *p* < 0.01, * *p* < 0.05, ^##^
*p* < 0.01, ^#^
*p* < 0.05. The figure marked ** was compared with the normal control group and the figure marked ## was compared with the Ang-II-induced-model group.

**Figure 5 marinedrugs-19-00655-f005:**
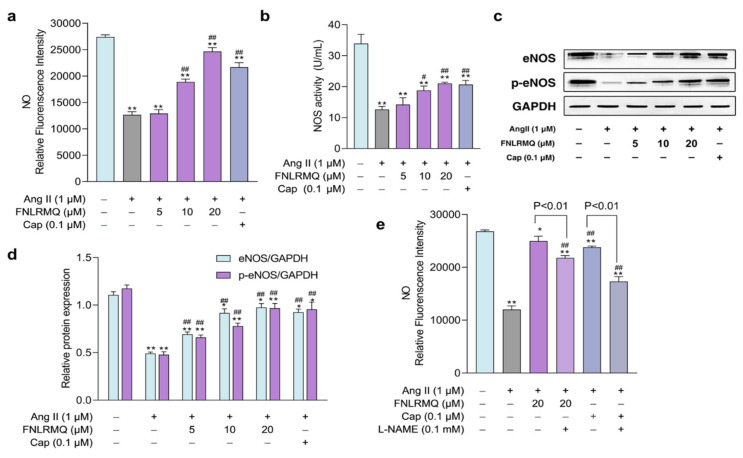
The effects of FNLRMQ on the production of NO and NOS and the role of the eNOS in response to Ang-II-induced oxidative stress. (**a**) The effects of FNLRMQ on the production of NO. HUVECs cultured with Ang II (1 µM) were treated with FNLRMQ (5–20 µM) for 24 h. Subsequently, the level of NO was determined with the fluorescent probe DAF-FM DA, following detection by a microplate reader. (**b**) NOS was detected by assay kits. (**c**) Western blotting analysis was performed to evaluate the effect of FNLRMQ on the expression of eNOS and p-eNOS. (**d**) The quantification of the immunoreactive bands. (**e**) HUVECs were initially exposed to the eNOS inhibitor L-NAME (100 μM) for 30 min, then treatment with Ang II and FNLRMQ for 24 h. Finally, the production of NO was detected. The values are expressed as the means ± SD, n = 3. ** *p* < 0.01, * *p* < 0.05, ^##^
*p* < 0.01, ^#^
*p* < 0.05. The figure marked ** was compared with the normal control group and the figure marked ## was compared with the Ang-II-induced-model group.

**Figure 6 marinedrugs-19-00655-f006:**
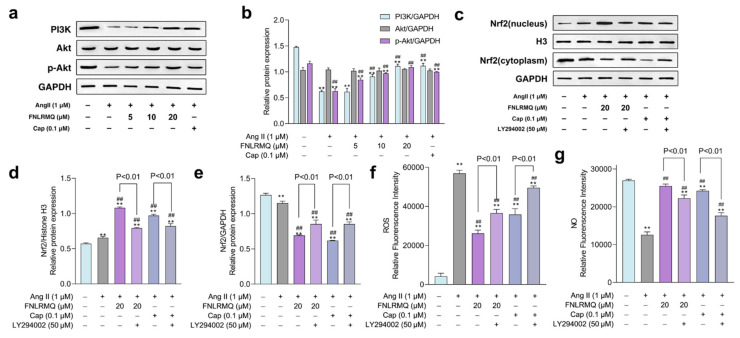
FNLRMQ attenuated Ang-II-induced HUVECs dysfunction through the PI3K/Akt pathway. (**a**) HUVECs treated with Ang II (1 µM) were incubated with FNLRMQ (5-20 µM) and Cap (0.1 μM) for 24 h, then the protein expression of PI3K, Akt, and p-Akt was detected by Western blotting analysis. (**b**)The quantification of the immunoreactive bands. (**c**) HUVECs were exposed to LY294002 (50 μM) for 30 min before being treated with Ang II for 24 h in the absence or presence of FNLRMQ. Subsequently, the expression of Nrf2 in the cytoplasm and nucleus was assessed by Western blotting analysis. (**d**,**e**) The quantification of the immunoreactive bands of Nrf2 in the nucleus and cytoplasm, respectively. (**f**,**g**) HUVECs were exposed to LY294002 (50 μM) for 30 min before being treated with Ang II for 24 h in the absence or presence of FNLRMQ (20 μM). Subsequently, the levels of ROS and NO were measured. The values are expressed as the means ± SD, n = 3. ** *p* < 0.01, ^##^
*p* < 0.01. The figure marked ** was compared with the normal control group and the figure marked ## was compared with the Ang-II-induced-model group.

## Data Availability

Not applicable.

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
