# Peer review of "ACE Inhibitory Peptide from Skin Collagen Hydrolysate of Takifugu bimaculatus as Potential for Protecting HUVECs Injury"

_marinedrugs, 2021, doi:10.3390/md19120655_

Round 1
Reviewer 1 Report
Reviewer’s report:
Manuscript entitled: “Angiotensin-I Converting Enzyme (ACE) Inhibitory Peptide 2 (FNLRMQ) Derived from Skin Collagen Hydrolysate of Pufferfish (Takifugu bimaculatus) as Potential for Protecting Human Umbilical Vein Endothelial Cells (HUVECs) Injury”.
The ACE inhibitory activity of a peptide from Takifugu bimaculatus (T. bimaculatus) is reported in the titled manuscript combining in vitro and in silico studies.
In the opinion of this reviewer the manuscript might be of interest but suffers in its presentation. Moreover, the biological and computational parts are strongly unpaired with the former well developed and the latter which is very weak more or less a scholarly exercise. This latter must be totally re-done.
Overall major modifications are requested:
- In order to make easier the reading of the manuscript its presentation needs to be improved. See for instance the “Screening and Identification of ACE Inhibitory Peptides” subparagraph consists of both experimental and in silico results.
- As above mentioned, the in silico results are inadequate. The computational work needs a deeper investigation and refinement for what is concerned with results and discussion: molecular docking is a powerful tool able to predict the binding affinity between two molecules and interpret/rationalize experimental results. In this work, the authors just list docking scores without providing the right interpretation of the in silico data. They do not provide any description about the binding mode of the active FNLRMQ into the ACE binding site.
- In the method section, authors did not report where (the search space) molecular docking was performed. A brief description of ACE structure could be provided.
Reviewer 2 Report
This research paper reports the identification of Angiotensin-I Converting Enzyme (ACE) Inhibitory Peptide (FNLRMQ) Derived from Skin Collagen Hydrolysate of Pufferfish (Takifugu bimaculatus) as Potential for Protecting Human Umbilical Vein Endothelial Cells (HUVECs) Injury. This paper presents interesting scientific data. I suggest to accept after minor corrections.
However, I found some minor grammatical mistakes and sentence formation issues. It should be revised thoroughly before acceptance.
Minor comments
Title: title may be shortened.
Abstract: Line no. 20: Delete the word "decreased"
Introduction: Line no. 71: write ..........and antihypertensive activities.
Results and discussion: Line no. 88: change it "they remain no bioactivity"
Results and discussion: Line no. 94: delete "respectively".
Results and discussion: Line no. 95: rewrite the sentence properly "The most potent with the highest"
Results and discussion: Line nos. 101-102: rewrite the sentence appropriately "FNLRMQ exhibited that 101 the highest ACE inhibition was 80.35%".
Results and discussion: Line no. 105: not "Unsurprisingly" it is "surprisingly"
Results and discussion: Line no. 136: what vascular?
Reviewer 3 Report
This paper reports an interesting pipeline for the discovery of bioactive peptides from skin collagen hydrolysate of pufferfish. Fish skin of farmed Takifugu bimaculatus is the major waste material of fish processing and valorization of by-products of food processing chain is of great importance in a circular economy perspective. The collagen from Takifugu bimaculatus was subjected to alkaline enzymatic hydrolysis, and the obtained peptide mixture was fractionated and analyzed by LC MS. Obtained 82 peptide sequences were evaluated in silico for their binding potential to angiotensin-I-converting enzyme (ACE). 22 best scored candidates were synthetized. The most potent oligopeptide FNLRMQ was evaluated in a multiparametric study using Ang II-induced HUVECs as an endothelial injury model. As concerning the chemical part, all relative experimental data are missing: 1) fractionation: HPLC traces, HRESI MS and MS/MS data of 22 peptides; 2) FNLRMQ peptide: HRESI MS and MS/MS of the synthetic sample and any experimental data (i.e. NMR spectra) proving its constitution and purity. As concerning the Pharmacological part, the work is professional in quality and the mechanism of action is well elucidated and circumstantiated, although it should be noted that, starting from inhibitory activity on the enzyme to all disclosed effects, comparable effects were observed at concentrations 100-1000 higher than the reference compound captopril.Author Response
Please see the attachment.
Thank you.

Round 2
Reviewer 1 Report
Reviewer’s report:
Manuscript entitled: “ACE Inhibitory Peptide from Skin Collagen Hydrolysate of Takifugu bimaculatus as Potential for Protecting HUVECs In- 3 jury”.
Minor modifications are requested:
Pag. 3 line 16 and 21: “amido” has to be corrected in “amide”
Pag. 3 line 19: “E377” has to be corrected in “D377”
Pag. 3 line 17, 18 and 19: “forms salt bridge” has to be corrected in “forms a salt bridge”
Pag. 3 line 21: “, forms ion contact” has to be corrected in “interacts”
Pag. 3 line 22: “Docking simulation studies indicates…..” has to be deleted.
Paragraph 3.4: “The MOE software was used to prepare docking…” has to be corrected in “The MOE software was used to perform molecular docking. 3D structures were minimized and the protonation state was checked [67]”
Caption of Figure S4:
“The detail binding model…” has to be corrected in “The detailed binding mode..”
…….. “zinc ion is depicted as magenta…” has to be corrected in “zinc ion is depicted as a sphere (magenta)” ……
“The blue dashes line are depicted as ion contacts, the red dashes line are depicted as salt bridges, and the orange dashes line are depicted as hydrogen bond interaction” has to be corrected as “H bonds, salt bridges and interactions with Zn++ are shown as orange, red and blue dashes lines”.
Author Response
RE: Revision Requested for marinedrugs-1458977
Manuscript ID: marinedrugs-1458977
Title: "ACE Inhibitory Peptide from Skin Collagen Hydrolysate of Takifugu bimaculatus as Potential for Protecting HUVECs Injury"
Journal: Marine Drugs
Reviewer(s)' Comments to Author:
Reviewer: 1
Manuscript entitled: “ACE Inhibitory Peptide from Skin Collagen Hydrolysate of Takifugu bimaculatus as Potential for Protecting HUVECs Injury”.
Reply: Thank you for the reviewer’s careful reading and valuable comments.
Minor modifications are requested:
Pag. 3 line 16 and 21: “amido” has to be corrected in “amide”
Pag. 3 line 19: “E377” has to be corrected in “D377”
Pag. 3 line 17, 18 and 19: “forms salt bridge” has to be corrected in “forms a salt bridge”
Pag. 3 line 21: “, forms ion contact” has to be corrected in “interacts”
Pag. 3 line 22: “Docking simulation studies indicates…..” has to be deleted.
Paragraph 3.4: “The MOE software was used to prepare docking…” has to be corrected in “The MOE software was used to perform molecular docking. 3D structures were minimized and the protonation state was checked [67]”
Caption of Figure S4:
“The detail binding model…” has to be corrected in “The detailed binding mode..”
…….. “zinc ion is depicted as magenta…” has to be corrected in “zinc ion is depicted as a sphere (magenta)” ……
“The blue dashes line are depicted as ion contacts, the red dashes line are depicted as salt bridges, and the orange dashes line are depicted as hydrogen bond interaction” has to be corrected as “H bonds, salt bridges and interactions with Zn++ are shown as orange, red and blue dashes lines”.
Reply: Many thanks the reviewer for your careful observation. We have already corrected all the typos that you mentioned in main body and ESI.
Thank you very much for considering our work and for the constructive comments. We hope that this new version is satisfactory.
Reviewer 3 Report
The revisions are sufficient and the manuscript is now acceptable for publication.
Author Response
RE: Revision Requested for marinedrugs-1458977
Manuscript ID: marinedrugs-1458977
Title: "ACE Inhibitory Peptide from Skin Collagen Hydrolysate of Takifugu bimaculatus as Potential for Protecting HUVECs Injury"
Journal: Marine Drugs
Reviewer(s)' Comments to Author:
Reviewer: 3
The revisions are sufficient and the manuscript is now acceptable for publication.
Reply: We are very grateful for the reviewer’s approval.